# Lifecycle of *Dryocosmus kuriphilus* Yasumatsu and Diversity and Importance of the Native Parasitoid Community Recruited in the Northern Region of Portugal

**DOI:** 10.3390/insects15010022

**Published:** 2024-01-01

**Authors:** Ana Lobo Santos, Sonia Alexandra Paiva Santos, Pedro António Casquero, Albino Bento

**Affiliations:** 1Centro Nacional de Competências dos Frutos Secos, 5300-263 Bragança, Portugal; bento@ipb.pt; 2Escuela de Doctorado, Universidad de Léon, 24071 León, Spain; 3Instituto Politécnico de Setúbal, Escola Superior de Tecnologia do Barreiro, Rua Américo da Silva Marinho, 2839-001 Lavradio, Portugal; sonia.santos@estbarreiro.ips.pt; 4Grupo Universitario de Investigación en Ingeniería y Agricultura Sostenible (GUIIAS), Instituto de Medio Ambiente, Recursos Naturales y Biodiversidad, Universidad de León, Avenida Portugal 41, 24071 León, Spain; pacasl@unileon.es; 5Centro de Investigação de Montanha, Escola Superior Agrária, Instituto Politécnico de Bragança, Campus Sta. Apolónia, 5300-253 Bragança, Portugal

**Keywords:** *Castanea*, *Dryocosmus kuriphilus*, native parasitoids, parasitism rates

## Abstract

**Simple Summary:**

*Dryocosmus kuriphilus* Yasumatsu, considered one of the most harmful organisms for plants of the *Castanea* genus, induces the formation of galls on the buds and leaves, causing a reduction in branch growth and fruiting, which can drastically reduce the production and quality of chestnuts, leading to a decline in chestnut trees. This pest was first recorded in Portugal in 2014. The objective of this work was to learn the life cycle of the pest and the diversity of native parasitoids in the northern region of Portugal between 2017 and 2019.

**Abstract:**

In this work, the objective was to learn the life cycle of *D. kuriphilus* and the diversity of native parasitoids in the northern region of Portugal between 2017 and 2019. The places studied belonged to the regions of Entre-Douro-e-Minho, Beira Interior, and Trás-os-Montes. To achieve the proposed objectives, buds were collected for egg and larva observation, galls were collected for larva, pupa, and adult observation and monitoring, and emergency boxes were used to identify the fauna present in the galls. In this study, 92% of *D. kuriphilus* adults emerged between June and July, with emergences occurring until September. We also obtained adults in winter, so it is important to study, in future works, the hypothesis of this pest performing diapause. Regarding the study of native parasitoids, compared to other countries, the same families emerged, with good rates of natural parasitism, although with fluctuations over the years. In the three municipalities under study, 11 species were identified (*Eupelmus azureus* Ratzeburg, *Eupelmus urozonus* Dalman, *Eurytoma brunniventris* Ratzeburg, *Megastigmus dorsalis* (Fabricius), *Ormyrus pomaceus* (Geoffroy), *Sycophila iracemae* Nieves Aldrey, *Sycophila variegata* (Curtis), *Sycophila biguttata* (Swederus), *Torymus flavipes* (Walker), *Torymus auratus* (Mueller), and *Torymus notatus* (Walker)). The average parasitism rates varied between 1.92% and 10.68%.

## 1. Introduction

The European chestnut tree (Fagaceae, *Castanea sativa* Mill.) is an economically and socially important crop in Portugal, especially in the mountain areas located in the northern part of the country. Portugal produces an average (last five years) of 32,450.6 tons of chestnuts per year [1], with around 85% of national production located in the northern region, and it is the fifth producer of chestnuts in the world. China is the world’s largest producer (1,634,532.304 tons), followed by Turkey, Republic of Korea, and Italy [2].

The chestnut gall wasp, *Dryocosmus kuriphilus* Yasumatsu (Hymenoptera: Cynipidae), is a pest, native to China, that attacks species of the genus *Castanea*, and it has been established in Japan, North America, and Europe [3]. *Dryocosmus kuriphilus* larvae cause galls that grow on the buds, leaves, and petioles, weakening trees and affecting chestnut production [4]. In Portugal, this invasive pest was observed for the first time in May 2014, in the northwest, in the Minho region, and it is now distributed throughout all the chestnut-producing regions. Chestnut orchards affected by this pest can reach production losses of around 80% in the fourth year of the attack, which constitutes a threat for the sustainability of the chestnut sector [5].

The chestnut gall wasp is a thelytokous parthenogenic (i.e., it produces diploid females from unfertilized eggs) and univoltine (i.e., it produces one generation per year) insect, whose life cycle is synchronized with the chestnut tree phenology; that is, the development of *D. kuriphilus* follows the activity of the chestnut tree. During the dormant period (from mid-November to early April) of the tree, the larvae hibernate in the chestnut buds; when the tree restarts its activity, with the development of leaves, this pest continues its development, promoting the appearance of galls [3,6]. Across the world, it has devastated chestnut production, with yield losses between 60% and 80% and an overall decline in chestnut tree health [3,7]. Plant galls protect the larval stages from predation or parasitism, but some of the individuals can experience high levels of mortality inflicted by a species-rich community of insect natural enemies [8]. In all invaded areas, *D. kuriphilus* has been parasitized by native parasitoid wasps associated with oak galls, with local variations in the species composition that depend on phenological overlaps and habitat features [9].

In China, the chestnut gall wasp populations were parasitized by a complex of natural enemies, including eleven species from five families belonging to the Chalcidoidea superfamily. Among them, the most successful species was *Torymus sinensis* Kamijo (Hymenoptera: Torymidae), which is a univoltine larval parasitoid, highly host-specific, and phenologically adapted to the life cycle of *D. kuriphilus* [3].

In Europe, since 2012, six hymenopteran families (Eulophidae, Eupelmidae, Eurytomidae, Ormyridae, Pteromalidae, and Torymidae) have been collected from *D. kuriphilus* galls. However, native European parasitoid species usually show low parasitism rates due to their low level of specialization [3,10].

The host–parasitoid dynamic is not influenced by the characteristics of the chestnut tree (variety, management system, etc.), although variables such as the climate or sampling date affected the natural parasitism rates [11].

According to the study developed in Calabria (Italy) by Bonsignore et al. [11], native parasitoids have little ability to control the chestnut gall wasp, largely due to the fact that 50% of native parasitoid species emerged before the first emergence of adult *D. kuriphilus*. The native parasitoid species were active and followed a temporal succession, and the parasitoid species with longer ovipositors (*Torymus auratus* Mueller and *Megastimus dorsalis* (Fabricius)) parasitized the chestnut gall wasp later, emerging during the last week of June and during the month of July [11]. *Dryocosmus kuriphilus* adults in the pharate phase were also parasitized, and some species acted as hyperparasitoids (*Mesopolobus* spp.) [11].

Natural parasitism, as mentioned above, is influenced by numerous variables, one of which is the landscape where the chestnut orchard is located. A more diverse habitat can support alternative hosts and food for native parasitoids, providing a variety of microhabitats that increase the parasitism and reproductive success of these species. According to some studies, higher parasitism rates were recorded in habitats with higher landscape diversity [11,12,13,14].

Considering the importance of native parasitoid species in limiting *D. kuriphilus* populations, the present study aims at evaluating the life cycle of *D. kuriphilus*, the diversity of native parasitoid species in northern Portugal, and the rates of natural parasitism. This knowledge can give us clues about the type of management practices that can be used to increase those species in the agroecosystem.

## 2. Materials and Methods

### 2.1. Study Areas

This study was carried out in three different regions located in the north of Portugal (Figure 1): Entre-Douro-e-Minho, Beira Interior, and Trás-os-Montes (Figure 1). These regions were chosen because they have relevant areas of chestnut cultivation and differ in terms of the management of chestnut orchards (Table 1). Climatically, they are relatively different.

The climate in Entre-Douro-e-Minho and Beira Interior is classified as a temperate climate, but with a mild and dry summer (Entre-Douro-e-Minho: annual mean precipitation of 1465.7 mm, annual mean temperature ranging from 9 °C to 20 °C; Beira Interior: annual mean precipitation of 882 mm, annual mean temperature ranging from 7 °C to 14.7 °C). The climate in Trás-os-Montes is classified as a temperate climate with a hot and dry summer (annual mean precipitation of 758.3 mm, annual mean temperature ranging from 6.7 °C to 17.9°C) (climate normal values for the period 1971–2000 from IPMA—Portuguese Institute of Meteorology and Atmosphere) [15].

In the Entre-Douro-e-Minho region, the sampling area was located in the municipality of Barcelos (B), where *D. kuriphilus* appeared for the first time in Portugal, in 2014. This region had small and very dispersed mature chestnut orchards and had a severe level of infestation (Table 2). The chestnut orchard located in B1 was surrounded by a forest of oaks (*Quercus pyrenaica* Willd.).

In the Beira Interior region, sampling was carried out in the municipality of Trancoso (T). In this region, the chestnut gall wasp was detected for the first time in 2014, and the adult’s chestnut groves have a severe level of infestation (Table 2).

In the Trás-os-Montes region, the study of the diversity of parasitoids was carried out in the municipality of Vinhais (V). In this region, the chestnut gall wasp was detected for the first time in 2015. In the sites selected for this study, the chestnut trees were adults and they had a severe level of infestation (Table 2).

### 2.2. Dryocosmus kuriphilus Yasumatsu Life Cycle

The study of the life cycle of *D. kuriphilus* was carried out in the Trás-os-Montes region in Vinhais (V7), and 10 buds (inactive phase of the cycle) or galls (active phase of the cycle) were randomly collected from 10 chestnut trees to evaluate the development of immature states over time (eggs, larvae, and pupae). Buds or galls were hand collected (at a 2.0 ± 0.3 m height) on a weekly basis from October to May (the active phase of the cycle) and on a monthly basis from May to October. In the laboratory, immediately after harvest, buds or galls were dissected and observed using a binocular microscope (model S9E LEICA) to assess the developmental stage of *D. kuriphilus*. The emergence of adults was assessed through the emergence boxes, and the date of eclosion of the adults was recorded. For this, *D. kuriphilus* galls were collected in June 2019, before the emergence of parasitoids. Six orchards were sampled in each region, and 240 galls were randomly collected in each orchard and transported to the laboratory. Galls were gently brushed to remove all debris, and 120 galls were transferred to each emergency box (Figure 2), giving a total of 12 emergency boxes per region. The boxes were made of cardboard (32 cm × 27 cm × 10 cm) with two holes, where two tubes (3 cm in diameter and 12 cm in length) were inserted and functioned as collecting tubes for emerged insects. Half of the emergency boxes were placed in a covered outdoor space, to simulate the weather conditions in the chestnut orchard, and the tubes were replaced every time they contained insects.

### 2.3. Sampling of D. kuriphilus Natural Parasitoid Communities

*D. kuriphilus* galls were collected in June from 2017 to 2019, before the emergence of parasitoids, according to the methodology previously described.

In October, and again in May, each box was examined in the laboratory, and emerged insects were removed and placed in microtubes (1.5 mL) that were frozen for further identification. Parasitoids were identified with the collaboration of the entomologist Jose Luis Nieves-Aldrey (Museo Nacional de Ciencias Naturales, Madrid—chalcidoid expert), and an identification key (unpublished) provided by the author [16].

In 2019, the day after the gall harvest (June), half of the emergency boxes were placed in a covered outdoor space, to simulate the weather conditions in the chestnut orchard, and the tubes (50 mL) were replaced every time they contained insects. The other half were monitored following the previous methodology. This methodology allowed the temporal evaluation of the emergence of *D. kuriphilus* parasitoids.

### 2.4. Rate of D. kuriphilus Natural Parasitism

In August from 2017 to 2019, *D. kuriphilus* galls were collected from chestnut trees in the Entre-Douro-e-Minho, Beira Interior, and Trás-os-Montes regions.

In each region, four chestnut orchards were selected (B1, B2, B3, B4; T1, T2, T3, T4; V1, V2, V3, V4), and 100 galls were randomly collected by hand from 10 trees (10 galls/tree) in each orchard at around 1.8 m to 2.0 m height. The galls were taken to the laboratory in paper bags in a cooler and were counted and dissected to evaluate the number of pupation chambers using a binocular microscope (model S9E LEICA Wetzlar, Germany) (Figure 3). Native parasitoids were counted from the emergency boxes used to identify the parasitoid community associated with galls in northern Portugal. The parasitism rate (%) was calculated using Equation (1):(1)$$\mathrm{Rate}\;\mathrm{of}\;\mathrm{natural}\;\mathrm{parasitism}\left(\%\right)=\frac{\mathrm{No}.\;\mathrm{of}\;\mathrm{parasitoid}\;\mathrm{sampled}\;}{\mathrm{total}\;\mathrm{number}\;\mathrm{of}\;\mathrm{chambers}\;}\times 100$$

### 2.5. Data Analysis

The life cycle of *D. kuriphilus* was shown using the proportion of the number of individuals over the months of the year and subsequent conversion into a relative percentage.

Parasitoid abundance (N), species richness (S), and diversity (D) data were combined per region and year and analyzed according to the variables of region, year, and region × year interaction. For this statistical analysis, the R software version 3.5.1 [17] was used. General linear models (GLMs) were performed to compare the abundance and richness of native parasitoids. All variables were analyzed after checking their error distributions and compliance with homoscedasticity. For diversity, an analysis of variance (ANOVA) was performed to compare year and interaction between year and regions.

Diversity was calculated based on Simpson’s Diversity Index (D) using the Equation (2):(2)$$D=\;\sum_{i=1}^{s}p_{i}^{2}$$
where $p_{i}^{2}$ is the proportion of individuals of the *i*th species, and *S* is the total number of species.

Rates of natural parasitism were compared based on the average natural parasitism rate, and the year and local variables were used. An ANOVA was carried out, and we performed Tukey’s pairwise comparison, Levene’s test for homogeneity of variance, and a Welch F test in the case of unequal variances. The statistical program used was PAST version 2.05. Significance levels for all analyses were set at *p* < 0.05.

## 3. Results

### 3.1. Dryocosmus kuriphilus Yasumatsu Life Cycle

According to the evaluation of the buds and galls harvested in the field, the life cycle of *D. kuriphilus* (Figure 4) occurred throughout a year (Figure 5). The peak of adults occurred from June to September, but 0.1% of the adults emerged in February.

Regarding eggs, 99% of all eggs were abundant in October. In the case of first-instar larvae, 60% were present from November to January. In April, 65% of second-instar larvae were emerged and, in May, 65% of third-instar larvae were emerged.

As for the pupae, the three pupal instar developed in June and July. The first instar of development is concentrated in June (95%). Between June and July, 86% of the pupae appear in the second instar of development, 93% of the pupae in the third instar of development, and 92% of adults.

### 3.2. D. kuriphilus Natural Parasitoid Communities

Twelve *D. kuriphilus* parasitoid species were identified, belonging to five families: Eurytomidae, Eupelmidae, Torymidae, Pteromalidae, and Ormyridae (Table 3).

During the 3 years of study, the following species emerged from galls collected: *Eupelmus azureus* Ratzeburg, *Eupelmus urozonus* Dalman, *Eurytoma brunniventris* Ratzeburg, *M. dorsalis*, *Ormyrus pomaceus* (Geoffroy), *Sycophila biguttata* (Swederus), *Sycophila iracemae* Nieves Aldrey, *Sycophila variegata* (Curtis), *T. auratus*, *Eurytoma setigera* (pistacina) Mayr, *Torymus flavipes* (Walker), and *Torymus notatus* (Walker). The genus *Mesopolobus* sp. occurred with some expression, but it was not possible to identify the species using our key.

The abundance of native parasitoids did not show significant differences (*p* > 0.05) between the variables of location and year (Figure 6A).

Figure 6A: same small letters on bars indicate no significant differences between years; A—same capital letters on bars indicate no significant differences between years by location (*p* > 0.05);

Figure 6B: same small letters on bars indicate significantly different values between locations; b—no significant differences between locations; A—same capital letters on bars indicate no significant differences between years by location.

Figure 6C: same small letters on bars indicate significantly different values between years; b—significantly different between years; ab—no significant differences between years; A—same capital letters on bars indicate no significant differences between years by location.

Regarding the species richness of native parasitoids, the region of Trás-os-Montes was significantly different from the regions of Entre-Douro-e-Minho and Beira Interior (*p* < 0.05). There were no significant differences between years (*p* > 0.05), or between the region of Entre-Douro-e-Minho and Beira Interior (Figure 6B).

Regarding species diversity, there were significant differences (*p* < 0.05) between 2017 and 2019. Between 2018 and the other two years, there were no significant differences (*p* > 0.05). As for the locations, there were no significant differences (*p* > 0.05) between them (Figure 6C).

In relation to the places with greater diversity and greater abundance, they changed over the years (Table 4).

During the study of native parasitoids present in northern Portugal, it was possible to conclude that the species present did not differ between regions, but their abundance varied throughout the year (2019) (Figure 7).

*Mesopolobus* sp. dominated in the three municipalities under study, followed by *O. pomaceus* in the Trás-os-Montes and Beira Interior regions, and *S. iracemae* and *S. variegata* in the Entre-Douro-e-Minho region.

Native parasitoids mostly emerged between the second half of June and the month of August (Figure 7).

In addition to the native parasitoids, individuals of the introduced species *T. sinensis* appeared in our samples. In Barcelos, in 2019, 19.6% of the parasitoids collected were *T. sinensis*; in Trancoso, in 2018, 16.5% and in 2019, 14.8% were *T. sinensis*; finally, in Vinhais in 2018, 4.2% were *T. sinensis* and in 2019, only 1.2% were *T. sinensis*. This parasitoid was introduced in Portugal as a means of biological control against *D. kuriphilus* in 2015 [5].

### 3.3. Rates of Natural Parasitism

The mean rates of natural parasitism varied between 1.92% and 10.68% over the study period (Figure 8).

The Entre-Douro-e-Minho region presented rates between 0.93% and 18.73% between the years 2017 and 2019. There were no significant differences between years (*p* > 0.05).

Regarding the Beira Interior region, the rates varied between 2.09% and 31.00% from 2017 to 2019. There were no significant differences between years (*p* > 0.05).

For the Trás-os-Montes region, the rates varied between 0.11% and 16.04% in the years 2017, 2018, and 2019. There were no significant differences between years (*p* > 0.05).

In relation to the locations, there were significant differences between T4 and B1, B2, V2 and V3 (*p* < 0.05).

## 4. Discussion

The life cycle of *D. kuriphilus* obtained in this study showed that some adults started to emerge in winter, February, which may indicate that, under certain conditions, the diapause could be broken earlier than that was described by Bernardo et al. [18]. As part of the work monitoring the releases of *T. sinensis* in the municipalities of Bragança and Vinhais, 15 individuals of *D. kuriphilus* emerged in 3 years of monitoring in galls collected in January (unpublished data).

The flight period of *D. kuriphilus*, in this study, occurred from the end of May in Minho (27 May 2020) to September in Trancoso (5 September 2019) and Vinhais (6 September 2019) but, in the bibliography, we found that adult emergence normally peaked between the middle of June and July [18].

Larvae had three stages, which developed throughout the year until they evolved into pupae. In the northern region of Portugal, there were three periods of the greatest concentration of each of them, L1 in November/January, L2 in April, and L3 in May. These dates coincide with the periods identified by Viggiane and Nugnes (2010) [19]. In the study carried out by these researchers, the following periods were identified, L1 between July/August and March/April, L2 in April/May for one month, and L3 at the end of April/May [19].

The different instars of development of the pupae are distributed between the months of June and July, coexisting with the adults of *D. kuriphilus* [20].

The natural parasitoid communities that emerged from *D. kuriphilus* galls in northern Portugal showed some similarities to those identified in other European countries. In Italy, 20 chalcid parasitoid species were identified [21], 15 were identified in Croatia, and 16 in Slovenia [22]. The most common *D. kuriphilus* parasitoid species identified in Europe were: *E. azureus*; *E. urozonus*; *E. brunniventris*; *S. variegata*; *O. pomaceus*; *Mesopolobus sericeus* Förster; *Mesopolobus tibialis* Westwood; *T. auratus*; *T. flavipes*; *Torymus geranii* (Walker); and *M. dorsalis* [23,24,25,26,27,28].

Outside Europe, the recruitment of these species of native parasitoids has also been verified; 15 parasitoid species were identified from galls collected in surveys performed in several Chinese provinces, 21 in Japan, and 12 in Korea [22].

*Eupelmus azureus* was found in two regions (Entre-Douro-e-Minho and Beira Interior) in Portugal, and has also appeared in Italy, Slovenia, and Hungary. In China, Japan, Korea, Italy, and Hungary, *E. uruzonus*, *E. brunniventris*, and *O. pomaceus* were found. *Megastigmus dorsalis* occurred in Italy and Hungary, and *S. biguttata* and *S. iracemae* only occurred in Italy. *Sycophila variegata* was referenced in China and Italy. In the USA and Italy, *T. auratus* and *T. flavipes* were referenced [22]. The chestnut orchard located in V2 was surrounded by a forest of oaks, a host plant for most of the native parasitoids present in the chestnut’s galls [11].

Regarding the diversity of species of native parasitoids that emerged in our study, there were no differences among locations.

In relation to abundance, despite the variation in species that appeared annually, there were no significant differences between years or study locations.

The Trás-os-Montes region showed lower species richness of native parasitoids. In the Entre-Douro-e-Minho and Beira Interior regions, the values were higher, but *D. kuriphilus* has been established for a longer time, and these are regions with a milder climate and greater humidity, which are generally more favorable to the development of parasitoids, recruiting a greater number of native parasitoids [28].

Based on the sampling carried out in the three regions (12,960 galls collected), it was possible to affirm that the native parasitoids were present, for the most part, during two generations and were not specific to *D. kuriphilus*; these characteristics may lower the effectiveness of pest control. The low level or low efficiency of natural enemies to which invaders might be susceptible offers a natural enemy escape opportunity [21]. In the Torymidae family, there were species that were specific, such as *T. sinensis*, which only parasitizes *D. kuriphilus.* But most species belonging to this family have several hosts, as is the case of *T. flavipes* (40 hosts, genus: *Andricus*, *Aphelonyx*, *Biorhiza*, *Chilaspis*, *Diplolepis*, *Cynips*, *Dryocosmus*, *Neuroterus*, *Plagiotrochus*, *Pseudoneuroterus*, and *Trigonaspis)*, *T. auratus* (49 hosts, genus: *Andricus*, *Aphelonyx*, *Biorhiza*, *Cerroneuroterus*, *Chilaspis*, *Cynips*, *Diastrophus*, *Dryocosmus*, *Neuroterus*, *Plagiotrochus*, *Pseudoneuroterus*, *Synophrus*, and *Trigonaspis*), and *T. notatus* (13 hosts, genus: *Andricus*, *Biortiza*, *Dryocosmus*, *Neuroterus*, *Plagiotrochus*, and *Synophrus*), which were found in Fagaceas and Rosaceas [29]. *Eupelmus urozonus*, *S. iracemae*, and *O. pomaceus* are generalist parasitoids that probably moved from oak to chestnut, parasitizing one more host [30].

The composition of native parasitoid communities and rates of parasitism can be related to the ground cover vegetation and flora, the altitude and the relative humidity [11,12,13,14]. The regions where we carry out our study are similar in terms of flora and climate, which translates into the emergence of the same species of native parasitoids.

Conservation biological control requires a thorough understanding of the mechanisms that allow native parasitoids to adapt and coexist with new hosts. This adaptation is fundamental for the success of the biological control of pests, namely, that of the chestnut gall wasp [11].

In the study carried out by Matošević and Melika (2013) [27], the locations that presented a greater diversity and abundance of native parasitoids were located near oak trees. In our case, the locations next to oak trees presented a different result; only B1 presented greater diversity in 2017 and 2019. In relation to V2, which also has oak trees in the surrounding area, it always had a low diversity and abundance.

During the study, 7200 galls were dissected, which made it possible to evaluate the number of chambers and calculate the rates of natural parasitism in the north of Portugal. The results exceeded initial expectations, as, according to other studies, this rate is lower in most countries. The bibliography refers to rates between 0.5% and 30%; in the north of Portugal, the rates are between 0.11% and 31%, but the majority are above 10% [26,27].

## 5. Conclusions

*Dryocosmus kuriphilus* adults mostly emerge between June and September, but 0.1% emerge in February. Knowledge of the life cycle of *D. kuriphilus* makes it possible to adapt the biological control of this pest, promoting the success of *T. sinensis* releases.

For the study of alternatives/complements to combat this pest, the life cycle is an essential tool.

The emergence of adults of *D. kuriphilus* in winter is a point to be explored in further studies.

This study allowed us to identify the species of native parasitoids that predominate in the north of Portugal, especially *O. pomaceus*, *S. iracemae*, and *S. variegata*, as well as the genus *Mesopolobus* sp. Twelve *D. kuriphilus* parasitoid species were identified, belonging to five families: Eurytomidae, Eupelmidae, Torymidae, Pteromalidae, and Ormyridae.

Comparing diversity and abundance in the sites sampled in this study, we observed that, despite there being sites with oak trees in the surrounding area, these did not consistently present higher values. In future work, it would be important to study the real influence of the presence of oaks on the native parasitoid community.

With the identification of native parasitoids present in the galls of *D. kuriphilus*, it is possible to create conditions that promote an increase in their presence, benefiting the growth of the rate of natural parasitism.

The mean rates of natural parasitism varied between 1.92% and 10.68% over the study period. The lowest natural parasitism rate was obtained in Vinhais (0.11%) in 2018, and the highest in Trancoso (31%) in 2017. Within each region, there is a large oscillation between years, namely, in Minho from 0.93% to 18.73%, in Beira Interior from 2.09% to 31%, and in Trás-os-Montes from 0.11% to 16.04%. Based on these results, we can state that native parasitoids are insufficient to combat *D. kuriphilus*. These parasitoids, as previously mentioned, are polyphagous, floating between different hosts. However, they are an important additional support in the biological fight against the parasitoid *T. sinensis*.

## Figures and Tables

**Figure 1 insects-15-00022-f001:**
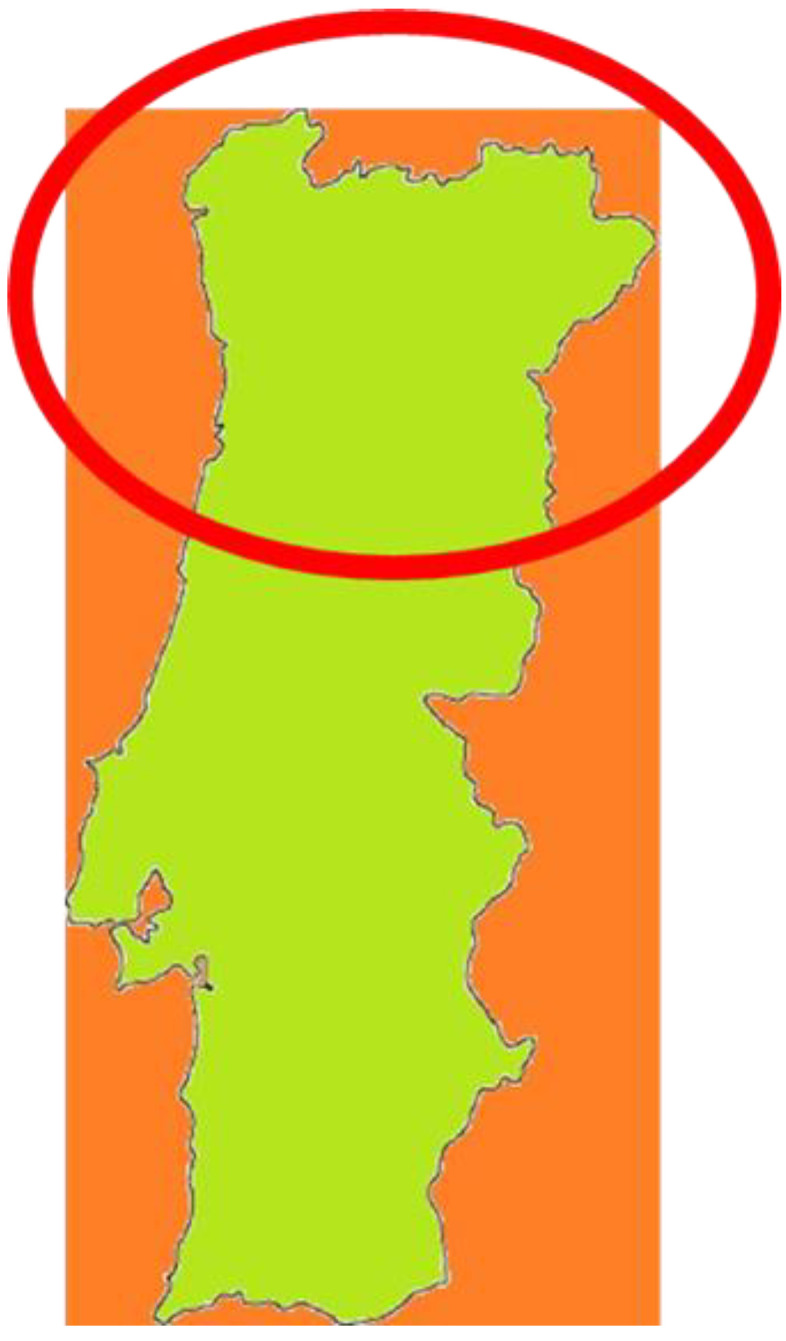
Map of the study area in Portugal.

**Figure 2 insects-15-00022-f002:**
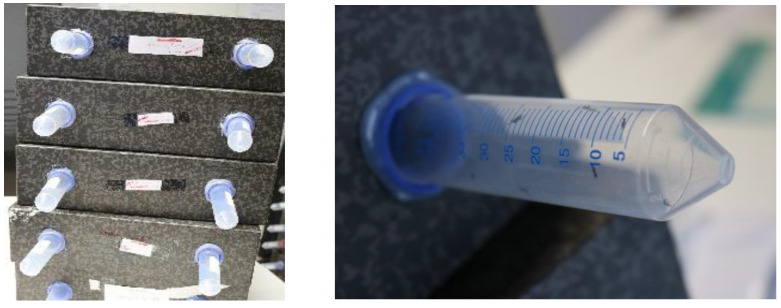
Emergency boxes used to collect parasitoids of *Dryocosmus kuriphilus* Yasumatsu.

**Figure 3 insects-15-00022-f003:**
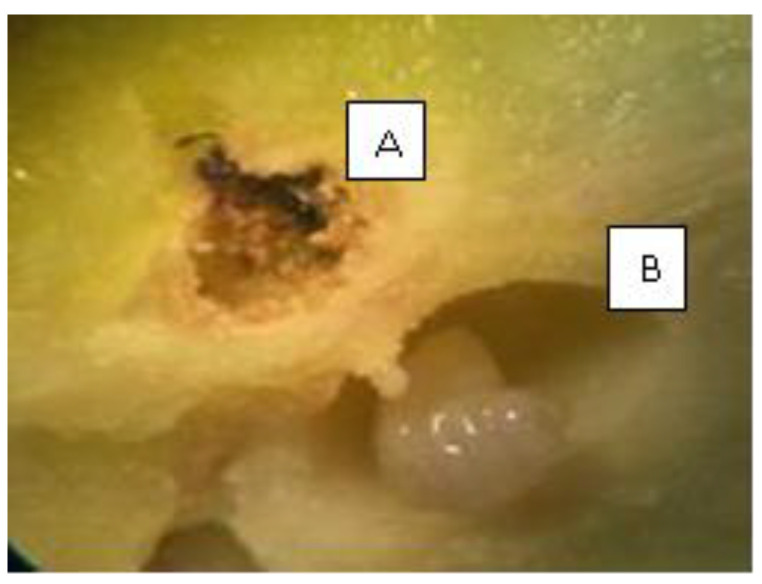
A—parasitoid chamber in a gall (parasitoid emerged); B—chamber of *Dryocosmus kuriphilus* Yasumatsu (presence of mature larvae).

**Figure 4 insects-15-00022-f004:**
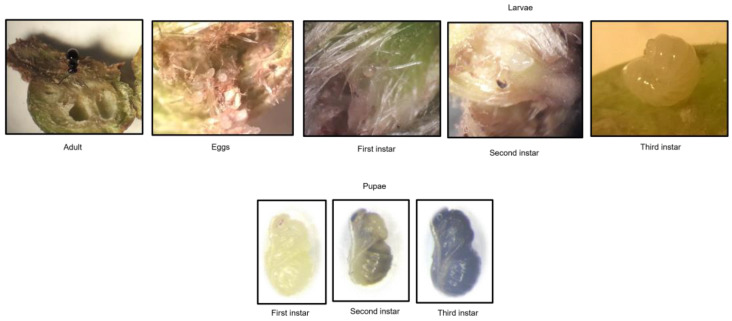
*Dryocosmus kuriphilus* Yasumatsu life cycle.

**Figure 5 insects-15-00022-f005:**
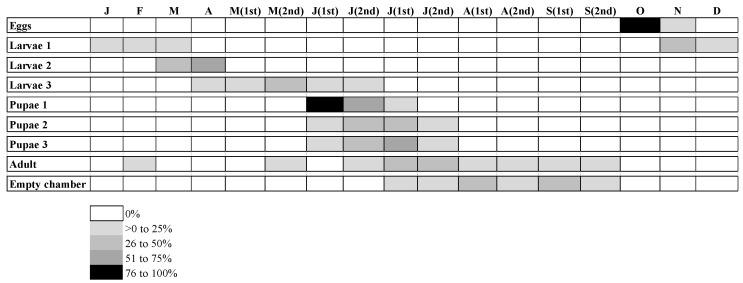
*Dryocosmus kuriphilus* Yasumatsu life cycle based on the evaluation of the buds and galls collected in the field throughout the year, in the north of Portugal (J—January; F—February; M—March; A—April; M (1st)—first half of May; M (2nd)—second half of May; J (1st)—first half of June; J (2nd)—second half of June; J (1st)—first half of July; J (2nd)—second half of July; A (1st)—first half of August; A (2nd)—second half of August; S (1st)—first half of September; S (2nd)—second half of September; O—October; N—November; D—December).

**Figure 6 insects-15-00022-f006:**
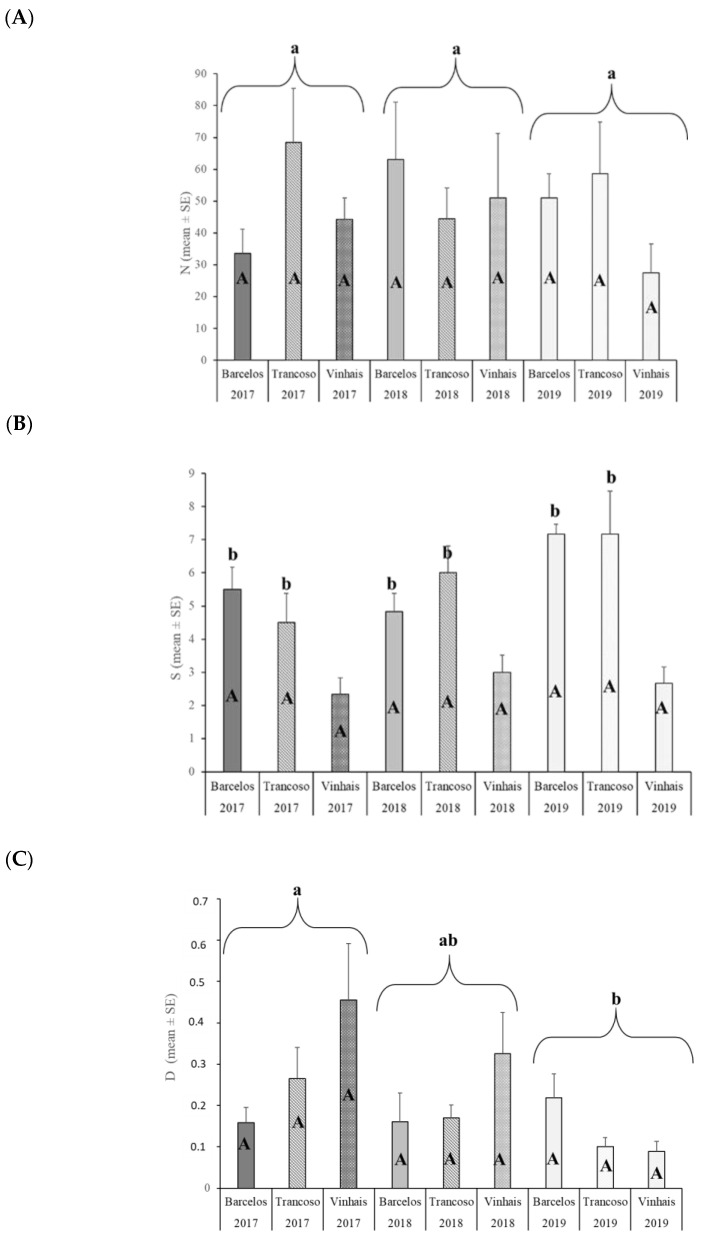
Abundance (mean ± SE) (**A**), species richness (mean ± SE) (**B**), and Simpson Diversity Index (mean ± SE) (**C**) of native parasitoids collected in each the 12 emergency boxes established for each sampling region (Entre-Douro-e-Minho: Barcelos, Beira Interior: Trancoso, and Trás-os-Montes: Vinhais) over three years (2017–2019).

**Figure 7 insects-15-00022-f007:**
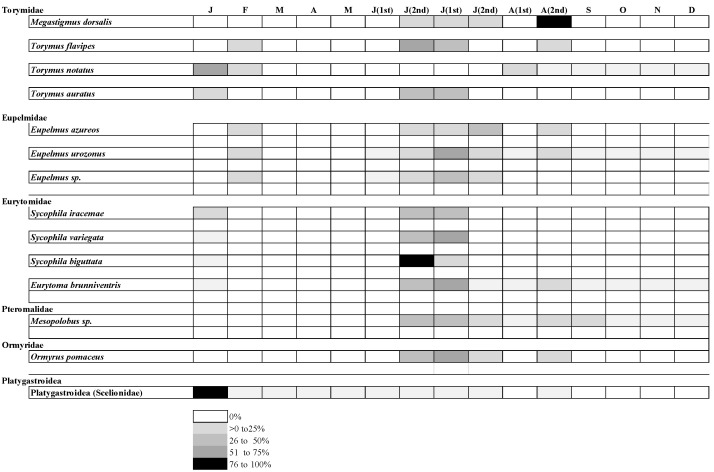
Relative abundance of native parasitoids throughout the year (J—January; F—February; M—March; A—April; M—May; J (1st)—first half of June; J (2nd)—second half of June; J (1st)—first half of July; J (2nd)—second half of July; A (1st)—first half of August; A (2nd)—second half of August; S—September; O—October; N—November; D—December).

**Figure 8 insects-15-00022-f008:**
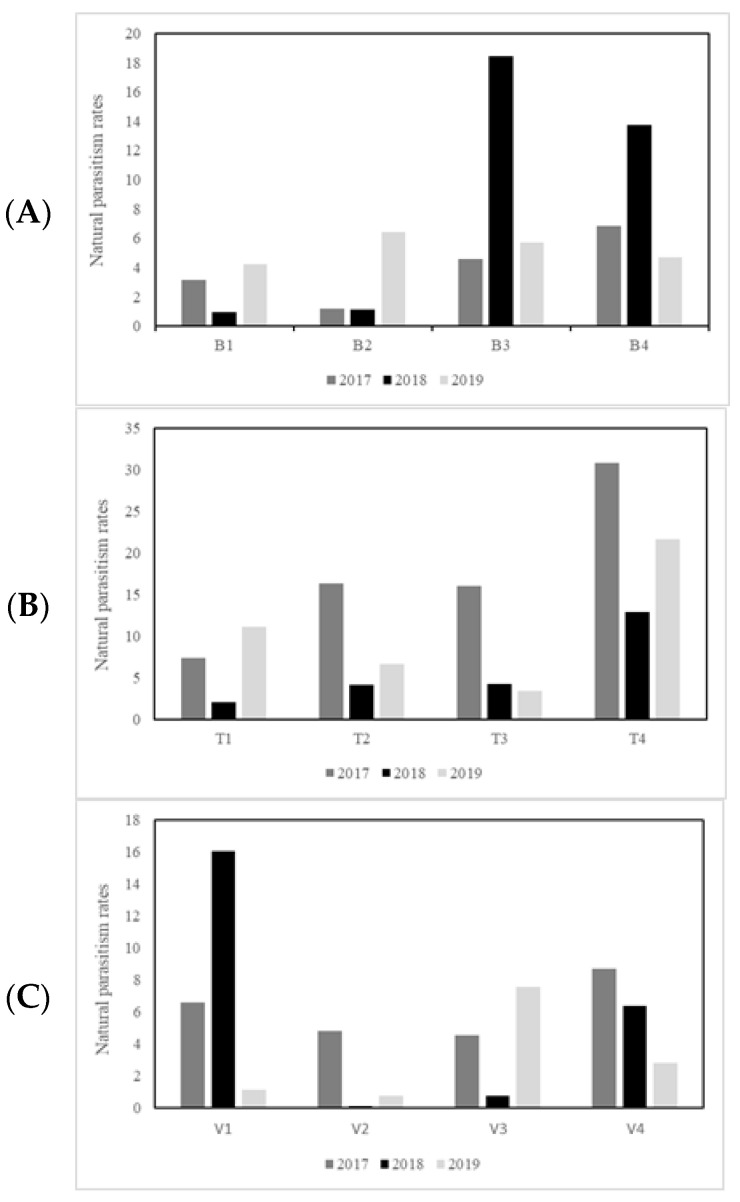
Rates of natural parasitism determined per year in the Entre-Douro-e-Minho (**A**), Beira Interior (**B**), and Trás-os-Montes (**C**) regions.

**Table 1 insects-15-00022-t001:** Characterization of the study areas in each region (Trás-os-Montes, Entre-Douro-e-Minho, and Beira Interior).

Region	Orchard	GPS Coordinates	Aim	Surrounding Area Dominated by	Soil Management
Trás-os-Montes	V1	41.850639 N–7.051500 W	Fruit production	Chestnut orchard	Tilled soil
	V2	41.859278 N–7.047250 W	Fruit production	Oaks	Tilled soil
	V3	41.899306 N–7.033056 W	Fruit production	Meadows	Tilled soil
	V4	41.887389 N–7.057583 W	Fruit production	Road	Tilled soil
	V5	41.854972 N–7.061778 W	Fruit production	Chestnut orchard	Tilled soil
	V6	41.813500 N–7.067139 W	Fruit production	Chestnut orchard	Tilled soil
	V7	41.82796 N–7.08440 W	Fruit production	Road	Tilled soil
Entre Douro e Minho	B1	41.568296 N–8.639190 W	Forest	Oaks	Underbrush
	B2	41.630517 N–8.654017 W	Fruit production	Road	Natural vegetation
	B3	41.626073 N–8.654342 W	Fruit production	Road	Natural vegetation
	B4	41.592550 N–8.620367 W	Fruit production	Road	Natural vegetation
	B5	41.631368 N–8.696132 W	Fruit production	Forest	Natural vegetation
	B6	41.549507 N–8.634450 W	Fruit production	Road	Natural vegetation
Beira Interior	T1	40.771164 N–7.364036 W	Forest	Chestnut orchard	Underbrush
	T2	40.769316 N–7.363713 W	Forest	Chestnut orchard	Underbrush
	T3	40.792497 N–7.333556 W	Fruit production	Road	Natural vegetation
	T4	40.792033 N–7.318378 W	Fruit production	Fresh fruit trees	Natural vegetation
	T5	40.679383 N–7.481569 W	Fruit production	Chestnut orchard	Natural vegetation
	T6	40.688978 N–7.476950 W	Production	Chestnut orchard	Natural vegetation

**Table 2 insects-15-00022-t002:** Infestation levels of *Dryocosmus kuriphilus* Yasumatsu [5]. The determination of the infestation level is based on visual observation of the entire canopy, in at least 3 to 5 trees, and on checking the presence of galls.

Level	Presence of Galls (%)	Classification
0	0–10	Initial infestation level
1	11–25	Low infestation level
2	26–50	Medium infestation level
3	51–80	Severe infestation level
4	>80	Very severe infestation level

**Table 3 insects-15-00022-t003:** Families and species of native parasitoids that emerged in the regions of Entre-Douro-e-Minho, Beira Interior, and Trás-os-Montes (Portugal) (relative%).

Family/Species		Locations		
		Entre-Douro-e-Minho	Beira Interior	Trás-os-Montes
**Eupelmidae**				
	*Eupelmus azureus*	2.00	0.43	0.00
	*Eupelmus urozonus*	9.34	7.68	4.77
**Torymidae**				
	*Megastigmus dorsalis*	3.11	4.17	1.54
	*Torymus auratus*	2.12	0.33	0.28
	*Torymus flavipes*	10.71	6.80	6.3
	*Torymus notatus*	0.87	0.11	0.00
**Eurytomidae**				
	*Eurytoma brunniventris*	3.61	6.47	0.84
	*Eurytoma setigera (pistacina)*	0.00	0.10	0.00
	*Sycophila biguttata*	0.50	1.21	0.14
	*Sycophila iracemae*	15.19	20.29	0.14
	*Sycophila variegata*	12.95	5.59	3.64
**Pteromalidae**				
	*Mesopolobus* sp.	28.89	27.96	45.24
**Ormyridae**				
	*Ormyrus pomaceus*	10.71	18.86	37.11
	**Total**	100	100	100

**Table 4 insects-15-00022-t004:** Places with greater diversity and greater abundance in each year.

Year	Region	Diversity	Abundance
2017	Entre-Douro-e-Minho	B1	B4
2017	Beira Interior	T5	T6
2017	Trás-os-Montes	V5	V6
2018	Entre-Douro-e-Minho	B4	B2
2018	Beira Interior	T4	T6
2018	Trás-os-Montes	V1	V1, V4, and V6
2019	Entre-Douro-e-Minho	B1	B6
2019	Beira Interior	T4	T6
2019	Trás-os-Montes	V4	V3

## Data Availability

The data presented in this study are available in the article.

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
