# Peer review of "Lifecycle of Dryocosmus kuriphilus Yasumatsu and Diversity and Importance of the Native Parasitoid Community Recruited in the Northern Region of Portugal"

_insects, 2024, doi:10.3390/insects15010022_

Round 1
Reviewer 1 Report
Comments and Suggestions for Authors
See attached file

Comments on the Quality of English Languagenone
Author Response
The authors express their gratitude to the reviewer for providing valuable comments and suggestions that helped enhance the quality of the manuscript.
Specific comments and suggestions:
Response: All the above suggestions have been inserted into the text. The resulting text changes are highlighted.
Others comments and suggestions:
Introdution:
Line 69 To say that the pest is “controlled” in China by a complex of 11 species is incorrect. The gall wasps is “atacked” or “parasitized” by those species, but most are generalist species that have liÆ©le or no impact on the pest’s density. Studies in Japan and Italy clearly show that a single species (Torymus sinensis) is the one the provides control (at least in those areas), as noted by the authors themselves on lines 71-72
Response: corrected throughout the document.
Methods:
Line 108 --What is a “soft and dry summer”? Do you mean “mild”?
Response: corrected throughout the document.
Table 1—The column labeled “soil management” uses classificaÆŸon (of soil management?) called “mobilized”, “underbrush” and “plant cover”, none of which seem like types of soil management.
Response: The name used is agronomically correct.
Line 122: “adult’s chestnut orchards”? Does this mean “mature chestnut orchards”?
Response: corrected throughout the document.
Line 151 “emergency boxes” should be “emergence boxes”
Response: corrected throughout the document.
Results:
Line 234-235-fig 3 -bottom of left column “camera” should be “chamber”
Response: corrected throughout the document.
Table 3. The use of punctuation in the table is confusing. Sometimes a period is used to demarcate the break between whole digits and decimal points, but in many other cases a comma is used. For English usage, all commas need to be changed to periods.
Response: corrected throughout the document.
Table 5. This table is of key importance and deserves greater emphasis. This species (T. sinensis) is the key parasitoid that is very likely (based on experience in Japan and Italy) going to control this pest. It was introduced into Portugal in 2015 and by 2019, it made up 36% of all parasitism (according to data present in Table 5). My prediction is that in a few years it will far exceed the atacks from the local parasitoids.
Response: The table was converted into text and percentages to highlight the information.
Discussion:
Lines 412 ff. “Conservation biological control requires a thorough understanding of the mechanisms that allow native parasitoids to adapt and coexist with new hosts. This adaptation is fundamental for the success of the biological control of pests, namely that of the chestnut gall wasp [11].” This passage suggests that local parasitoids are expected by the authors to become beÆ©er adapted to chestnut gall wasp and eventually control this pest. This statement cites as its support one study from southern Italy. In northern Italy, control of this pest followed the introduction of the Torymus sinensis and was not due to adaption by local parasitoids. The circumstances in southern Italy are unknown to me. Has T. sinensis been released there? In Japan, again, while a local parasitoid (T. beneficus) did provide considerable control, it was quickly displaced by T. sinensis, which provided better control after its introduction. So, I think this statement lacks support, as there is no clear case of local parasitoids providing adequate control. As such it is speculative at best and ignores substantial evidence to the contrary.
Response: The intention with this passage was not to predict the control of D. kuriphilus with native parasitoids. It was just pointed out that this type of biological control is time-consuming and requires in-depth knowledge of the life cycle of both.

Reviewer 2 Report
Comments and Suggestions for Authors
Manuscript ID: insects-2781957 entitled “Lifecycle of Dryocosmus kuriphilus Yasumatsu and diversity and importance of the native parasitoid community recruited in the North region of Portugal” by Ana Lobo Santos, Sónia A.P. Santos, Pedro A. Casquero, and Albino Bento submitted to section: Insect Pest and Vector Management, is an important study examined The Asian chestnut gall wasp life cycle and its extant parasitoids’ assembly in northern Portugal. I recommend consideration for publication in Insects MDPI pending minor revision. I added my comments in the attached PDF for authors’ revision. My main notes are:
· Figures 3,5: Explain in a footnote the months and why first and second samples.
· Explain in a footnote what is first stage, to third stage of pupa and what is larva 1, 2, 3.
· Needs a figure plate for the lifecycle showing good quality photos of eggs, larvae, and pupae. Now only figure 2 has a part of the lifecycle without a scale bar.
· Please add a figure plate of the most important parasitoids found during this study. This can enhance the quality of this article since all descriptions of the parasitoids are listed as unpublished keys. At least one parasitoid representing each of the five families.
· Table three total % parasitoids do not calculate to100% as reported. Please check the numbers.
· Figure 4 needs explanation for X and Y axis and the statistical grouping letters.
· Needs a map of Portugal showing Entre-Douro-e-Minho: Barcelos, Beira Interior: Trancoso and Trás-os-Montes: Vinhais.
· Mesopolobus sp. dominated in the three municipalities under study. Why was this important species not identified?
· Table 5. Proportion of Torymus sinensis in relation to the native parasitoids sampled. This table is not necessary. Authors can explain the content in two sentences. Besides, very little information on this parasitoid and its introduction to Portugal. This is the most important non-native parasitoid for D. kuriphilus. The parasitoid was detected from 2017 to 2019 in Barcelos, Trancoso and Vinhais. Reason for un-establishment in the study sites and history of its introduction and recovery in other sites should be discussed.

Author Response
The authors express their gratitude to the reviewer for providing valuable comments and suggestions that helped enhance the quality of the manuscript.
Specific comments and suggestions:
Response: All of the above suggestions have been inserted into the text. The resulting text changes are highlighted.
Figures 3,5: Explain in a footnote the months and why first and second samples. Explain in a footnote what is first stage, to third stage of pupa and what is larva 1, 2, 3. Needs a figure plate for the lifecycle showing good quality photos of eggs, larvae, and pupae. Now only figure 2 has a part of the lifecycle without a scale bar. Please add a figure plate of the most important parasitoids found during this study. This can enhance the quality of this article since all descriptions of the parasitoids are listed as unpublished keys. At least one parasitoid representing each of the five families.
Response: All of the above suggestions have been inserted into the text. The resulting text changes are highlighted.
Table 3. total % parasitoids do not calculate to100% as reported. Please check the numbers.
Response: corrected throughout the document.
Figure 4. needs explanation for X and Y axis and the statistical grouping letters.
Response: corrected throughout the document.
Needs a map of Portugal showing Entre-Douro-e-Minho: Barcelos, Beira Interior: Trancoso and Trás-os-Montes: Vinhais.
Response: include in the document.
Mesopolobus sp. dominated in the three municipalities under study. Why was this important species not identified?
Response: the species were not identified because with the keys available it wasn't possible and there was no funding to carry out molecular biology.
Table 5. Proportion of Torymus sinensis in relation to the native parasitoids sampled. This table is not necessary. Authors can explain the content in two sentences. Besides, very little information on this parasitoid and its introduction to Portugal. This is the most important non-native parasitoid for D. kuriphilus. The parasitoid was detected from 2017 to 2019 in Barcelos, Trancoso and Vinhais. Reason for un-establishment in the study sites and history of its introduction and recovery in other sites should be discussed.
Response: corrected throughout the document.
Others comments and suggestions:
Line 33-36, 41, 43, 48, 49, 56, 57, 59, 60,61, 63, 64, 72, 74, 77, 81, 85, 86, 87, 88, 94, 98, 99, 108-112, 124, 150, 151, 166, 167, 182, 190-191, 194, 200, 218, 253, 285, 286, 308, 317, 355, 358, 361, 363, 367, 373, 377, 379-381, 385, 393, 394, 398, 405, 410, 416, 420, 425-426, 431, 437, 452, 453, 460, 466, 478, 494, 502, 507, 510, 527
Response: corrected throughout the document.
Table 1., table 3 and table 4
Response: corrected throughout the document.
Line 45. Add (followed by Bolivia, Turkey, South Korea and Italy)
Response: Bolivia wasn't included because it doesn't produce castanea sativa.
Line 71, 130 and 168
Response: I don't understand what you want.
Line 153. Add de average size of orchards
Response: The size of the groves is not important because the groves are in continuous patches of chestnut trees, so I don’t have this information.
Line 240. Table 3 has 13 species in five families.
Response: There are 12 species and 1 genus, not 13 species.
Line 449. Within each region of Minho there is ….
Response: Changing it in the way you suggest changes the meaning of the sentence, the results are from 3 regions.
Table 5. Delete
Response: The table has been deleted and turned into a paragraph in the text.
